

**Impacts of an aerosol layer on a mid-latitude continental system of cumulus clouds:**
**how do these impacts depend on the vertical location of the aerosol layer?**
Seoung Soo Lee[1,2], Junshik Um[3,4], Won Jun Choi[5], Kyung-Ja Ha[2,4,6], Chang Hoon Jung[7],
Jianping Guo[8], Youtong Zheng[9]
[1]Earth System Science Interdisciplinary Center, University of Maryland, Maryland, USA
[2]Research Center for Climate Sciences, Pusan National University, Busan, Republic of
Korea
[3]Department of Atmospheric Sciences, Pusan National University, Busan, Republic of
Korea
[4]BK21 School of Earth and Environmental Systems, Pusan National University, Busan,
Republic of Korea
[5]National Institute of Environmental Research, Incheon, Republic of Korea
[6]Center for Climate Physics, Institute for Basic Science, Busan, Republic of Korea
[7]Department of Health Management, Kyungin Women's University, Incheon, Republic of
Korea
[8]State Key Laboratory of Severe Weather, Chinese Academy of Meteorological Sciences,
Beijing, China
[9]The Program in Atmospheric and Oceanic Sciences, Princeton University, Princeton,
New Jersey, USA



Corresponding authors: Seoung Soo Lee and Junshik Um
E-mail: cumulss@gmail.com, slee1247@umd.edu, jjunum@pusan.ac.kr













## 52  Abstract


Using the large-eddy simulation framework, effects of an aerosol layer on warm cumulus
clouds in the Korean Peninsula when the layer is above or around the cloud tops in the
upper atmosphere are examined. Also, these effects are compared to effects of an aerosol
layer when it is around or below the cloud bases in the low atmosphere. Simulations show
that when the aerosol layer is in the low atmosphere, aerosols absorb solar radiation and
radiatively heat up air enough to induce greater instability, stronger updrafts and more
cloud mass than when the layer is in the upper atmosphere. As aerosol concentrations in
the layer decrease, the aerosol radiative heating gets weaker to lead to less instability,
weaker updrafts and less cloud mass when the layer is in the low atmosphere. This in turn
makes differences in cloud mass, which are between a situation when the layer is in the
low atmosphere and that when the layer is in the upper atmosphere, smaller. It is found that
the transportation of aerosols by updrafts reduces aerosol concentrations in the aerosol
layer, which is in the low atmosphere, and in turn reduces the aerosol radiative heating,
updraft intensity and cloud mass. It is also found that the presence of aerosol impacts on
radiation suppresses updrafts and reduces clouds. Aerosols affect not only radiation but
also aerosol activation. In the absence of aerosol impacts on radiation, aerosol impacts on
the droplet nucleation increases cloud mass when the layer is in the low atmosphere as
compared to a situation when the layer is in the upper atmosphere. As aerosol impacts on
radiation team up with those on the droplet nucleation, differences in cloud mass, which
are between a situation when the layer in the low atmosphere and that when the layer is in
the upper atmosphere, get larger. This is as compared to a situation when there is no aerosol
impacts on radiation and only aerosol impacts on the droplet nucleation.









## 1. Introduction

Warm cumulus clouds play an important role in global hydrologic and energy circulations
(Warren et al., 1986; Stephens and Greenwald, 1991; Hartmann et al., 1992; Hahn and
Warren, 2007; Wood, 2012). With industrialization, there have been significant increases
in concentrations of aerosols acting as cloud condensation nuclei (CCN) and these
increases are known to decrease droplet size (Twomey, 1974, 1977). Increases in
concentrations of aerosols acting as radiation absorbers are also known to enhance the
radiative heating of air by aerosols. These aerosol-induced changes in droplet size and
radiative heating affect updrafts, cloud mass, cloud albedo and precipitation (Albrecht,
1989; Hansen et al., 1997). Global hydrologic and energy circulations are eventually
affected by these aerosol effects. However, these effects, which are particularly on warm
cumulus clouds, are highly uncertain and thus act to cause the highest uncertainty in the
prediction of future climate (Ramaswamy et al., 2001; Forster et al., 2007).
In recent years, people start to take interest in effects of aerosol layers above or around
the tops of clouds on clouds (e.g., de Graaf et al., 2014; Xu et al., 2017). This interest is
motivated by aerosol layers that are originated from biomass burning sites in the southern
Africa. These layers are lifted and transported to the southeast Atlantic (SEA) region and
located above or around the top of a large deck of warm cumulus and stratocumulus clouds
that play an important role in global hydrologic and energy circulations. Note that aerosols
in the transported aerosol layers contain organic carbon (OC) and black carbon (BC) that
act as radiation absorbers as well as cloud condensation nuclei (CCN). When these aerosols
act as radiation absorbers, they absorb solar radiation and heat up the atmosphere to change
atmospheric stability. This in turn affects the cumulus clouds, the hydrologic and energy
circulations, and climate. When these aerosols act as CCN, they have an impact on aerosol
activation, subsequent microphysical and precipitation processes in the cumulus clouds,
those circulations and climate. Reflecting the interest and an associated potential
importance of aerosol layers above or around cloud tops in the circulations and climate, to
better understand roles of aerosol layers above or around cloud tops in cloud development
and its impacts on climate, there were international field campaigns in the SEA such as the
NASA ObseRvations of Aerosols above CLouds and their intEractionS (ORACLES;





https://espo.nasa.gov/oracles/content/ORACLES), the UK Clouds and Aerosol Radiative
Impacts and Forcing (CLARIFY; Redemann et al., 2021) and the French Aerosol,
Radiation and Clouds in southern Africa (AEROCLO-sA; Formenti et al., 2019)
campaigns.
It is well-known that the relative vertical location of an aerosol layer and a cloud deck
can affect cloud properties (e.g., updrafts, cloud mass and albedo) that responds to aerosol
absorption, subsequent changes in atmospheric stability, aerosol activation and subsequent
changes in microphysics and precipitation (Haywood and Shine, 1997; Johnson et al., 2004;
McFarquhar and Wang, 2006). However, despite above-mentioned field campaigns,
previous studies on aerosol-cloud interactions have focused mainly on effects of aerosols
around or below cloud bottoms on clouds. Effects of aerosols above or around cloud tops
on clouds have not been examined as much. This contributes to the low-level understanding
of effects of the relative location of an aerosol layer and a cloud deck on the cloud deck.
Improving this understanding, which is about going beyond the traditional approach that
focuses on around- or below-cloud-bottom aerosol layers, is likely to contribute to the more
comprehensive understanding of aerosol-cloud-radiation interactions and thus more
general parameterizations of those interactions for climate models. Hence, this study aims
to enhance our understanding of effects of the relative location of an aerosol layer and a
cloud deck on the cloud deck. This aim is pursued by investigating aerosol-cloud-radiation
interactions in a typical situation where an aerosol layer is around or below the bottom of
a system of warm cumulus clouds. Then, to fulfill the aim, these interactions in the typical
situation are compared to a situation where an aerosol layer is around or above the top of
the system. In this study, the investigation is performed by using simulations adopting the
large-eddy simulation (LES) framework and an idealized setup, which is based on
observation, for the aerosol layer.

**2.  Case, model and simulations**

**2.1 LES model**




The Advanced Research Weather Research and Forecasting (ARW) model is for LES
simulations in this study. The ARW model is a compressible model with a nonhydrostatic
status. A 5th-order monotonic advection scheme is used to advect microphysical variables
(Wang et al., 2009). The ARW adopts a bin scheme to parameterize microphysics. The
Hebrew University Cloud Model (HUCM) detailed in Khain et al. (2011) is the bin scheme.
A set of kinetic equations is solved by the bin scheme to represent size distribution
functions for each class of hydrometeors and aerosols acting as cloud condensation nuclei
(CCN). The hydrometeor classes are water drops, ice crystals (plate, columnar and branch
types), snow aggregates, graupel and hail. There are 33 bins for each size distribution in a
way that the mass of a particle $m_j$ in the j bin is to be $m_j = 2m_{j-1}$.
Aerosol sinks and sources control the evolution of aerosol size distribution. These
sinks and sources include advection and aerosol activation (Fan et al., 2009). Activated
particles are emptied in the corresponding bins of the aerosol spectra. Aerosol mass
included in hydrometeors, after activation, is moved to different classes and sizes of
hydrometeors through collision-coalescence and removed from the atmosphere once
hydrometeors that contain aerosols reach the surface.
The Rapid Radiation Transfer Model (RRTM; Mlawer et al., 1997) has been coupled
to the bin microphysics scheme described above. Aerosols before their activation can affect
radiation by changing the reflection, scattering, and absorption of radiation. This radiative
effect of aerosol is represented following Feingold et al. (2005). The internal aerosol
mixture and the ARW relative humidity are used to calculate the hygroscopic growth of
the aerosol particles as well as their optical properties, represented by extinction, single
scattering albedo, and asymmetry factor. Aerosol uptake of water vapor is considered over
the range of relative humidity in the domain. In practice, hygroscopic growth and optical
property calculations are performed offline prior to simulation and stored in lookup tables.
Calculations are done for the prescribed aerosol size distribution and composition, and unit
concentration. During model runtime, grid-point number concentration and relative
humidity determine the look-up table entries that specify the grid-point aerosol optical
properties. The effective sizes of hydrometeors are calculated in an adopted microphysics
scheme and the calculated sizes are transferred to the RRTM to consider effects of the
effective sizes on radiation.



The presence of the aerosol perturbs the radiative fluxes reaching the surface, and its
subsequent partitioning into sensible and latent heat fluxes (i.e., the Bowen ratio). This is
accounted for with the interactive Noah land surface model (Chen and Dudhia, 2001).

**2.2 Case and simulations**

**2.2.1 Case and standard simulations**

There is an observed system of warm cumulus clouds in a domain in the Korean Peninsula,
as marked in Figure 1, for a period between 10:00 and 18:00 LST on April 13th, 2015. This
system is simulated to fulfill the goal of this study. For a three-dimensional simulation (i.e.,
the control run) of the system in the domain over the period, a 50-m resolution is used for
the horizontal domain. The length of the domain in the east-west (north-south) direction is
20 (20) km. In the vertical domain, the resolution coarsens with height. The resolution in
the vertical domain is 20 m just above the surface and 100 m at the model top that is at ~
4.5 km in altitude. Initial and boundary conditions of potential temperature, specific
humidity, and wind for the simulation are provided by reanalysis data. These data are
produced by the Met Office Unified Model (Brown et al., 2012) every 6 hours on a 0.11°
× 0.11° grid. These data represent the synoptic-scale environment. Figure 2 depicts the
vertical distributions of potential temperature and water-vapor mixing ratio at 09:00 LST
on April 15th, 2015 in radiosonde sounding that is obtained near the domain. This vertical
distribution represents initial environmental conditions for the development of the clouds
in the control run. The conditional instability is present in the vertical profiles and this
favors the development of warm cumulus clouds. An open lateral boundary condition is
employed for the control run.
There is a site of the aerosol robotic network (AERONET; Holben et al., 2001) in the
domain. At 10:00 LST when clouds in the domain start to develop, there is an aerosol layer
advected from the East Asia and this layer causes aerosol pollution in the domain. This
advection is monitored by aerosol-measuring stations in the Yellow sea as described in Lee
et al. (2021). According to the AERONET measurement, which is 1 hour before the
observed cumulus clouds start to form, aerosol particles in the layer, on average, are an



internal mixture of 70 % ammonium sulfate, 22 % organic compound and 8% black carbon.
Aerosol chemical composition in this study is assumed to be represented by this mixture
in the whole domain during the whole simulation period. Based on the AERONET
observation, the size distribution of background aerosols acting as CCN is assumed to
follow a bi-modal log-normal distribution. Modal radius of this distribution is 0.11 and 1.2
μm and standard deviation of this distribution is 1.71 and 1.92, while the partition of
aerosol number, which is normalized by the total aerosol number of the size distribution,
is 0.999 and 0.001 for accumulation and coarse modes, respectively. It is assumed that the
size distribution of background aerosols acting as CCN in all parts of the domain during
the whole simulation period is assumed to follow the bi-modal size distribution. The
average aerosol concentration in the layer over the domain at 10:00 LST is ~15000 cm$^{-3}$.
This average concentration is applied to all grid points in the layer at the first time step of
the control run. This aerosol layer is idealized to be located around and below cloud bases
between the surface and 1.0 km as shown in Figure 3a. Cloud bases are located around 1
km. Above the layer, aerosol concentration is assumed to be 150 cm$^{-3}$.
This study aims to understand differences in aerosol effects on warm cumulus clouds
between the situation where an aerosol layer is above and around the cloud tops and that
where the aerosol layer is around and below the cloud bases. To fulfill this goal, we repeat
the control run by an idealized setup where the aerosol layer is moved upward to altitudes
between 2.5 and 3.5 km as shown in Figure 3b. Altitudes between 2.5 and 3.5 km are places
where cloud tops are located frequently. Note that the simulated maximum cloud-top height
is 3.3 km. This repeated run is referred to as the aro-above-cld run. As shown in Figures
3a and 3b, aerosol concentrations in the aerosol layer are 15000 cm$^{-3}$ in both of the runs.
Outside the main aerosol layer, aerosol concentration is set at 150 cm$^{-3}$ in both of the runs
(Figures 3a and 3b). Here, we see that the depth of the main aerosol layer and aerosol
concentrations in the main layer are identical between the runs.
It is well-known that aerosol-cloud-radiation interactions are strongly dependent on
aerosol concentrations (Tao et al., 2012). We want to test how results in the control and
aro-above-cld runs are sensitive to aerosol concentrations in the main aerosol layer. For
the test, the control and aro-above-cld runs are repeated with 10 times lower aerosol
concentrations in the main aerosol layer but with no changes in aerosol concentrations in



the domain outside the main aerosol layer. In these repeated runs, the aerosol concentration
in the main aerosol layer at the first time step is 1500 cm$^{-3}$. Henceforth, the repeated control
and aro-above-cld runs are referred to as the control-1500 and aro-above-cld-1500 runs.

**2.2.2 Additional simulations**

Clouds affect aerosols by cloud processes such as nucleation of droplets and aerosol
transportation (or advection) by cloud-induced wind. Updrafts and downdrafts comprise
cloud-induced wind and transport aerosols upward and downward, respectively. Here, we
are interested in impacts of clouds on aerosols and how these impacts in turn affect aerosol
effects on clouds. To examine these aspects of aerosol-cloud interactions, the above-
mentioned four standard simulations (e.g., the control, aro-above-cld, control-1500 and
aro-above-cld-1500 runs) are repeated by preventing aerosol evolution with time at each
grid point. In other words, in these repeated runs, aerosol concentrations at each grid point,
which are set at the first time step, do not vary with time or are not affected by cloud
processes such as nucleation and advection. These repeated runs are referred to as the
control-novary, aro-above-cld-novary, control-1500-novary, and aro-above-cld-1500-
novary runs. By comparing the standard simulations to these repeated ones, roles of cloud
impacts on aerosols in aerosol-layer impacts on clouds are identified.
In this study, we aim to better understand roles of the interception (e.g., reflection,
scattering and absorption) of radiation by aerosols, which results in phenomena such as
radiative heating of air by aerosols. These roles are referred to as aerosol radiative effects.
To better understand aerosol radiative effects, the above four standard simulations are
repeated again by turning off aerosol radiative effects. These repeated runs are the control-
norad, aro-above-cld-norad, control-1500-norad, aro-above-cld-1500-norad runs. The
summary of simulations in this study is given in Table 1.

**3. Results**

**3.1 The control and aro-above-cld runs**



Figure 4 shows the time- and area-averaged vertical distributions of cloud-liquid mass
density that represents cloud mass for the standard simulations. In Figure 4, the cloud layer
is between 1.0 and 3.3 km in the control run and between 0.8 and 2.6 km in the aro-above-
cld run. The time- and domain-averaged cloud-liquid mass density is 0.7 and $1.3 \times 10^{-3}$ g
$m^{-3}$ in the control run and in the aro-above-cld run, respectively. Hence, we see that clouds
are thicker with their higher tops and have greater mass in the control run than in the aro-
above-cld run. This is despite the fact that aerosol concentrations in the main aerosol layer
in the control run are identical to those in the aro-above-cld run.

Figure 5a shows the time series of the domain-averaged liquid-water path, which is

the vertical integral of cloud-liquid mass density and thus, represents cloud mass, for the
standard simulations. During the initial stage of the cloud development between 12:50 and
13:50 LST, the average cloud mass is slightly higher in the control run than in the aro-
above-cld run. Also, the average non-zero cloud mass starts to appear earlier in the control
run than in the aro-above-cld run. Over the period between 13:50 and 14:10 LST, there is
a jump (or rapid increase or surge) in the average cloud mass in the control run but not in
the aro-above-cld run. During this period with the jump, at some specific time points, the
average mass is ~one order of magnitude higher in the control run than in the aro-above-
cld run. Of interest is that just after the jump and at 14:10 LST, the average mass in the
control run starts to decrease and at 14:40 LST, becomes lower than that in the aro-above-
cld run. Hence, the greater time- and domain-averaged cloud mass in the control run is
mainly attributed to the jump in cloud mass. As seen in Figures 5b and 5c that show the
time series of the domain-averaged updraft speed and condensation rates, respectively, the
average updraft mass fluxes and associated condensation rates in the control run are also
slightly higher in the control run than in the aro-above-cld run for the period between 12:50
and 13:50 LST. The average updraft speed and associated condensation rates in the control
run jump during the period between ~13:50 and ~14:10 LST, hence, these speed and rates
are much higher in the control run than in the aro-above-cld run during the period between
~13:50 and ~14:10 LST (Figures 5b and 5c). After the jump, the speed and rates decrease
rapidly and become lower in the control run than in the aro-above-cld run (Figures 5b and
5c). Taking into account the fact that condensation is the only source of cloud mass and
the updraft speed strongly control the amount of condensation, the updraft speed,



condensation rate and cloud mass in each of the runs and differences in those variables
between the runs are similar in terms of their temporal evolution.

Figure 5d shows the time series of the domain-averaged convective available potential

energy (CAPE) for the control and aro-above-cld runs. Considering that updrafts grow by
consuming buoyancy energy, updraft intensity is proportional to CAPE that is the integral
of the buoyancy energy in the vertical domain. Hence, the evolution of CAPE in each of
the runs is similar to that of the updraft speed, associated condensation rates and cloud
mass. Accordingly, the evolution of differences in CAPE between the runs is similar to that
of those differences in the updraft speed, associated condensation rates and cloud mass.
This similarity includes the jump not only in CAPE but also in those speed, rates and mass
during the period between 13:50 and 14:10 LST in the control run. (052022)

In Figures 5, the peaks (or the maximum values) of the domain-averaged CAPE,

updrafts, condensation rates and cloud mass in the control run occur around 14:10 LST and
this occurrence is earlier than that which occurs around 14:50 LST in the aro-above-cld
run. This means that the cloud system in the control run reaches its mature stage earlier
than that in the aro-above-cld run. After the peak around 14:10 LST, the system enters its
dissipating stage in the control run, while around 14:10 LST in the aro-above-cld run, the
system still evolves to enter the mature stage and it enters its dissipating stage after 14:50
LST.  Hence, the cloud system in the control run matures and demises faster as compared
to that in the aro-above-cld run.  Stated differently, the cloud system in the control run has
a shorter life cycle than that in the aro-above-cld run.

To find mechanisms controlling the jump in CAPE which acts as a main cause of the

greater cloud mass in the control run, the analysis of the results is done for an initial period
between 10:00 LST when the simulation starts and 13:50 LST which is immediately before
the jump starts to occur. The thermodynamic condition for the CAPE jump is established
during this initial period. The average net shortwave fluxes at the surface are shown in
Table 2 for the initial period in the control and aro-above-cld runs. Table 2 shows that
during the initial period in the control run, there is a smaller amount of shortwave radiation
that are incident on the surface in the control run than in the aro-above-cld run. The aerosol
layer intercepts solar radiation and reduces solar radiation which reaches the surface. In
spite of the fact that the depth of the main aerosol layer and aerosol concentrations in the





layer are identical between the runs, results here indicate that the aerosol layer in the low
atmosphere is more efficient in the interception of solar radiation than that in the upper
atmosphere. Due to the less solar radiation reaching the surface, the time- and area-
averaged net surface heat fluxes, which are the sum of the surface sensible and latent-heat
fluxes, become lower in the control run than in the aro-above-cld run during the initial
period (Table 2). Hence, the surface fluxes favor more instability or higher CAPE and
associated more intense subsequent updrafts and cloud mass in the aro-above-cld run than
in the control run.
The vertical distributions of the time- and domain-averaged radiative heating rates are
obtained for the initial period for each of the control and aro-above-cld runs. For the initial
period, the average radiative heating rate is much higher in the control run than in the aro-
above-cld run particularly at altitudes between 0.0 and ~ 1.0 km where cloud bases are
located (Figure 6a). This is associated with the fact that the main aerosol layer is located at
altitudes between 0.0 and 1.0 km in the control run. This more radiative heating in the low
atmosphere favors subsequent higher CAPE, which involves its jump in the control run, in
the control run than in the aro-above-cld run after the initial period. The average radiative
heating rate is higher in the aro-above-cld run than in the control run at altitudes between
~2.5 and 3.5 km. This is associated with the fact that the main aerosol layer is located at
altitudes between 2.5 and 3.5 km in the aro-above-cld run. However, this higher radiative
heating rate is in the upper part of the domain and tends to stabilize the atmosphere more
in the aro-above-cld run than in the control run. Thus, the higher radiative heating rate in
the aro-above-cld run contributes to lower CAPE, less intense updrafts and lower cloud
mass in the aro-above-cld run especially for the period when the jumps occur in the control
run.
Effects of greater radiative heating in the low atmosphere on CAPE outweigh those
effects of solar radiation which are incident on the surface and the associated surface heat
fluxes during the initial stage in the control run. This leads to more intense clouds with
more cloud mass for the rest of periods, in turn leading to the more time- and domain-
averaged cloud mass in the control run than in the aro-above-cld run.

**3.2 Comparisons between simulations with different aerosol concentrations**




With the lower concentration of aerosols in the main layer, there are much more solar
radiation reaching the surface and resultant higher surface fluxes in the control-1500 run
than in the control run and in the aro-above-cld-1500 run than in the aro-above-cld run
(Table 2). This makes CAPE higher in the control-1500 run than in the control run over
most of the simulation period except for the period between 13:50 and 14:20 LST during
which the jump in CAPE in the control run exists, and in the aro-above-cld-1500 run than
in the aro-above-cld run throughout the simulation period (Figure 5d). Then, there are
stronger updrafts and greater condensation rate and cloud-liquid mass density developing
in the control-1500 run than in the control run over most of the simulation period, which
are except for the period between 13:50 and 14:20 LST during which the jump in updrafts
and cloud-liquid mass in the control run exists, and in the aro-above-cld-1500 run than in
the aro-above-cld run throughout the simulation period (Figures 5a, 5b and 5c). This leads
to the greater time- and domain-averaged cloud-liquid mass density in the control-1500 run
than in the control run and in the aro-above-cld-1500 run than in the aro-above-cld run
(Figure 4). Regarding the control and control-1500 runs, this is despite the fact that aerosol
radiative heating in the main aerosol layer in the low atmosphere is higher due to higher
aerosol concentrations there in the control run than in the control-1500 run (Figure 6). In
Figure 4, it is seen that the time- and domain-averaged cloud-liquid mass in the aro-above-
cld-1500 run is higher than in the control run. This is due to more solar radiation reaching
the surface in the aro-above-cld-1500 run than in the control run and despite the fact that
aerosol concentrations and thus aerosol radiative heating in the low atmosphere is much
higher in the control run than in the aro-above-cld-1500 run (Table 2 and Figure 6). In
Figure 4, it is also seen that the time- and domain-averaged cloud-liquid mass in the
control-1500 run is higher than in the aro-above-cld run. This is due to higher aerosol
concentrations and more aerosol radiative heating in the low atmosphere (in the upper
atmosphere) in the control-1500 run (aro-above-cld run) than in the aro-above-cld run
(control-1500 run).
Similar to the situation between the control and aro-above-cld runs, there is less solar
radiation reaching the surface in the control-1500 run than in the aro-above-cld-1500 run
(Table 2). In association with this, there is the less surface heat fluxes in the control-1500



run than in the aro-above-cld-1500 run. This favors higher CAPE and more invigoration of
updrafts and associated convection in the aro-above-cld-1500 run than in the control-1500
run. However, overall, CAPE is higher in the control-1500 run than in the aro-above-cld-
1500 run (Figure 5d). This is because similar to the situation between the control and aro-
above-cld runs, more aerosols are in the low atmosphere in the control-1500 run than in
the aro-above-cld-1500 run. These more aerosols heat up the low atmosphere more and
increase the instability there more (Figure 6c). This induces increases in CAPE, which
compensates for decreases in CAPE due to the smaller amount of solar radiation reaching
the surface in the control-1500 run, and leads to overall higher CAPE in the control-1500
run than in the aro-above-cld-1500 run.
Associated with higher CAPE, there is greater cloud-liquid mass density in the control-
1500 run than in the aro-above-cld-1500 run, which is similar to the situation between the
control and aro-above-cld runs. However, differences in the mass density between these
repeated runs are smaller than those between the control and aro-above-cld runs (Figure 4).
As seen in Figure 5a which shows the time series of the domain-averaged cloud-liquid
mass density, the control-1500 run does not show a jump in the mass density unlike the
situation in the control run. This contributes to smaller differences in cloud-liquid mass
density between the control-1500 and aro-above-cld-1500 runs than between the control
and aro-above-cld runs. The CAPE evolution of the control-1500 and aro-above-cld-1500
runs show that there is no jump in CAPE and thus updrafts in the control-1500 run (Figure
5d). This mainly contributes to smaller differences in CAPE and updrafts, which in turn
contributes to smaller differences in cloud mass between the control-1500 and aro-above-
cld-1500 runs than between the control and aro-above-cld runs.
In addition, remember that the cloud system in the control run has a shorter life cycle
than in the aro-above-cld run. However, as seen in Figure 5, the cloud system the conrol-
1500 run has a similar life cycle to that in the aro-above-cld-1500 run. In the control run,
the instability or CAPE accumulates or increases rapidly to reach its peak, which forms the
jump, for a period between 13:50 and 14:10 LST, while in the control-1500 run, CAPE
increases gradually to reach its peak from ~12:00 LST to ~14:30 LST (Figure 5d). For a
period between ~14:10 and ~14:50 LST, CAPE reduces rapidly down back to the CAPE
value around ~13:00 LST in the control run, while CAPE decreases gradually and never



drops back to CAPE value at ~12:00 LST until the end of the simulation period in the
control-1500 run. This leads to the shorter life cycle or lifetime of the system not only in
the control run than in the aro-above-cld run but also in the control run than in the control-
1500 run. Here, we see that as aerosol concentration increases in the main aerosol layer in
the low atmosphere, the time scale of the accumulation and consumption of the instability
or convective energy gets shorter, leading to the shorter lifetime of the cloud system. When
aerosol concentration in the main aerosol layer in the low atmosphere is relatively low as
in the control-1500 run, the relatively long lifetime of the cloud system in the control-1500
run is similar to the lifetime in the aro-above-cld-1500 run. However, when aerosol
concentration in the main aerosol layer in the low atmosphere is relatively high as in the
control run, the relatively short lifetime of the cloud system in the control run is shorter
than the lifetime in the aro-above-cld run.
Comparisons among the above four standard simulations show that with increasing
aerosol concentrations in the main aerosol layer, there are decreases in solar radiation
reaching the surface due to the increasing interception of solar radiation by aerosols,
leading to decreases in cloud-liquid mass density, whether the main aerosol layer is in the
low atmosphere or in the upper atmosphere. Also, when there is the main aerosol layer in
the low atmosphere, there is radiative heating of air by aerosols around and below cloud
bases and this enables more instability, stronger updrafts and more cloud-liquid mass
despite less solar radiation reaching the surface than when there is the main aerosol layer
in the upper atmosphere, whether aerosol concentrations in the main layer is or low
(initially set at 1500 cm$^{-3}$) or high (initially set at 15000 cm$^{-3}$)
The increase in cloud-liquid mass from the aro-above-cld-1500 run to the control-1500
run is smaller than that from the aro-above-cld run to the control run. This means that with
increasing concentrations of aerosols, the effects of radiative heating of aerosols, which is
in the low atmosphere, on instability and cloud-liquid mass enhances. This enhancement
is closely linked to the jump in the CAPE and updrafts that appears when concentration of
aerosols in the low atmosphere is high. Stated differently, the jump in the CAPE does not
occur when concentration of aerosols in the low atmosphere is low, meaning that there is
a critical value of initial aerosol concentrations above which the jump occurs.



### 3.3 Comparisons between simulations with predicted and prescribed aerosol concentrations

Figure 7 shows the vertical distributions of aerosol concentrations, which are averaged over the horizontal domain and simulation period, for the standard and repeated runs with no temporal variation of aerosols (e.g., the control-novary, aro-above-cld-novary, control-1500-novary, and aro-above-cld-1500-novary runs). Comparisons between the control and control-novary runs (between the control-1500 and control-1500-novary runs) show that due to the upward transportation of aerosols by updrafts in the control (control-1500) run, aerosol concentrations in the main aerosol layer in the low atmosphere reduces and those in the air above the main aerosol layer increases. This is as compared to the situation in the control-novary (control-1500-novary) run where aerosols are assumed not to be affected by cloud-induced wind (Figures 7a and 7c). Note that the low atmosphere is where cloud-induced updrafts develop and grow, hence, the upward transportation of aerosols by them is dominant. Due to the higher concentration of aerosols in the low atmosphere between 0.0 and ~1.0 km, there is more radiative heating of air by aerosols in the control-novary run than in the control run and in the control-1500-novary run than in the control-1500 run in the low atmosphere.

Comparisons between the aro-above-cld and aro-above-cld-novary runs (between the aro-above-cld-1500 and aro-above-cld-1500-novary runs) show that due to the transportation of aerosols by downdrafts in the aro-above-cld (aro-above-cld-1500) run, aerosol concentrations in the main aerosol layer in the upper atmosphere reduces and those in the air below the main aerosol layer increases. This is as compared to the situation in the aro-above-cld-noary (aro-above-cld-1500-novary) run where aerosols are assumed not to be affected by cloud-induced wind (Figures 7b and 7d). Note that the upper atmosphere is where cloud-induced updrafts decelerate and turn into downdrafts, and the downward transportation of aerosols by them is dominant. However, those increases in aerosol concentrations in the air below the main aerosol layer mainly occur for the atmosphere between ~1.5 and ~2.5 km and aerosol concentrations in the low atmosphere between 0.0 and ~1.0 km do not change significantly (Figures 7b and 7d). Hence, these transported aerosols by downdrafts do not affect instability in the low atmosphere, which tends to have



more impacts on CAPE than instability in other parts of the atmosphere, significantly. This
leads to similar instability in the low atmosphere and CAPE, which in turn leads to similar
updrafts and cloud mass between the aro-above-cld and aro-above-cld-novary runs and the
aro-above-cld-1500 and aro-above-cld-1500-novary runs (Figure 8a). However, due to
more aerosols and their more radiative heating of air in the low atmosphere in the control-
novary run than in the control run and in the control-1500-novary run than in the control-
1500 run, there are higher CAPE (or greater instability), stronger updrafts and higher cloud
mass in the control-novary run than in the control run and in the control-1500-novary run
than in the control-1500 run (Figure 8a). It is notable that cloud mass in the control-novary
run is so large that its maximum value in the vertical profile as shown in Figure 8a exceeds
that even in the control-1500-novary run (Figure 8a). Associated with this, there are only
~20 % changes in cloud mass between the control-1500 and control-1500-novary runs,
while there are as much as ~200 % changes in cloud mass between the control and control-
novary runs. This indicates that as aerosol concentration increases in the low atmosphere,
the sensitivity of responses of cloud mass to changes in aerosol concentrations, which are
induced by cloud-induced wind, in the low atmosphere increases substantially.

**3.4 Comparisons between simulations with aerosol radiative effects and those with**

**no aerosol radiative effects**


Figure 8b shows that with no aerosol radiative effects, differences in cloud mass between
the control-norad and aro-above-cld-norad runs are much smaller than those differences
between the control and aro-above-cld runs with aerosol radiative effects. However, as in
the control and aro-above-cld runs, there is higher cloud mass in the control-norad run,
which has the main aerosol layer in the low atmosphere, than in the aro-above-cld-norad
run, which has the main aerosol layer in the upper atmosphere. Figure 8b shows that cloud
mass in each of the control and aro-above-cld runs increases significantly when aerosol
radiative effects are turned off. Figure 8b shows that with no aerosol radiative effects,
differences in cloud mass between the control-1500-norad and aro-above-cld-1500-norad
runs are also smaller than those differences between the control-1500 and aro-above-cld-
1500 runs with aerosol radiative effects. However, as in the control-1500 and aro-above-



cld-1500 runs, there is higher cloud mass in the control-1500-norad run, which has the
main aerosol layer in the low atmosphere, than in the aro-above-cld-norad run, which has
the main aerosol layer in the upper atmosphere (Figure 8b). Figure 8b shows that cloud
mass in each of the control-1500 and aro-above-cld-1500 runs increases when aerosol
radiative effects are turned off, although these increases are a lot smaller than those in each
of the control and aro-above-cld runs. Here, we see that aerosol radiative effects suppress
clouds and reduce cloud mass. This means that effects of aerosol-induced reduction in the
surface-reading solar radiation and subsequently in the surface heat fluxes on clouds are
dominant over those of radiative heating of air by aerosols, when it comes to the
explanation of differences in cloud mass between a situation with aerosol radiative effects
and that with no aerosol radiative effects. The suppression of clouds and reduction in cloud
mass increase with increasing aerosol concentrations in the main aerosol layer, whether the
main layer is in the low atmosphere or in the upper atmosphere, since more aerosols reduce
the surface-reaching solar radiation and surface heat fluxes more.

More aerosols and their activation (or nucleation of droplets) produce higher cloud

droplet number concentration (CDNC) in the low atmosphere in the control-1500-norad
run than in the aro-above-cld-1500-norad run. Note that aerosol activation mainly occurs
around cloud bases which are located in the low atmosphere. Droplets act as a source of
condensation, since individual droplets provide their surface areas onto which water vapor
condenses.  Hence, higher CDNC induces more condensation and this in turn induces
stronger updrafts and more cloud mass in the control-1500-norad run than in the aro-above-
cld-1500-norad run. These effects of more aerosols, which induces more condensation and
stronger updrafts, are generally referred to as aerosol microphysical effects (Lee et al.,
2016). The differences in CDNC are greater due to greater differences in aerosols in the
low atmosphere between the control-norad and aro-above-cld-norad runs than those
between the control-1500-norad and aro-above-cld-1500-norad runs. This leads to greater
aerosol microphysical effects or greater differences in condensation, associated updrafts
and cloud mass between the control-norad and aro-above-cld-norad runs than those
between the control-1500-norad and aro-above-cld-1500-norad runs.  With aerosol
radiative effects, radiative heating of air in the low atmosphere works in tandem with
aerosol microphysical effects. Hence, it is shown that as compared to the situation with no



aerosol radiative effects, with aerosol radiative effects, differences in cloud mass between
the run with the main aerosol layer in the low atmosphere and that with the layer in the
upper atmosphere are greater, whether aerosol concentrations are low (initially set at 1500
$cm^{-3}$) or high (initial set at 15000 $cm^{-3}$) in the layer. In addition to the jump in CAPE,
updrafts and condensation in the control run as described in Section 3.1, greater aerosol
microphysical effects in the low atmosphere when aerosol concentrations are high (initial
set at 15000 $cm^{-3}$) in the aerosol layer contributes to greater differences in cloud mass
between the control and aro-above-cld runs than between the control-1500 and aro-above-
cld-1500 runs when aerosol concentrations are low (initial set at 1500 $cm^{-3}$) in the aerosol
layer.

The initial concentration of aerosols in the aro-above-cld-norad run is identical to that

in the aro-above-cld-1500-norad run in the low atmosphere where most of aerosol
activation occurs as seen in the description of initial aerosol distribution in Section 2.2.
Due to this, in the low atmosphere, CDNC and condensation in the aro-above-cld-norad
run are similar to those in the aro-above-cld-1500-norad run. This leads to similar cloud
mass between the runs.

**4. Summary and conclusions**

This study examined differential impacts of an aerosol layer on warm cumulus clouds in
the Korean Peninsula between a situation where the main aerosol layer is located around
or above the tops of clouds in the upper atmosphere and that where the main aerosol layer
is located around or below the bottoms of clouds in the low atmosphere. This study finds
that the main layer intercepts more solar radiation which reaches the surface when it is in
the low atmosphere than when it is in the upper atmosphere. This makes the surface heat
fluxes and associated CAPE lower, which tend to make updrafts weaker and make cloud
mass lower when the main aerosol layer is in the low atmosphere. However, with the main
aerosol layer in the low atmosphere, there is a greater amount of cloud mass than that with
the main layer in the upper atmosphere. This is because the main layer in the low
atmosphere heats up the air there more, leading to increases in the instability and CAPE as
compared to the situation when the main layer is in the upper atmosphere. These increases


in CAPE are larger than reduction in CAPE due to the reduced solar radiation reaching the
surface, resulting in more cloud mass when the main layer is in the low atmosphere than
when the main layer is in the upper atmosphere.

With decreasing concentrations of aerosols in the main aerosol layer, there are

decreases in the interception of solar radiation reaching the surface, increases in surface
heat fluxes, CAPE and cloud mass whether the main layer is in the low atmosphere or in
the upper atmosphere. However, the decreasing concentrations of aerosols cause the jump
in instability as seen in the evolution of CAPE to disappear when the main layer is in the
low atmosphere. This leads to reducing differences in cloud mass between the situation
with the main layer in the low atmosphere and that with the layer in the upper atmosphere.
When the main aerosol layer is in the low atmosphere, with increasing aerosol
concentrations in the layer, the lifetime of cloud system reduces in a way that the lifetime
with the main layer in the low atmosphere gets shorter than that with the layer in the upper
atmosphere.

Updrafts and downdrafts in clouds transport aerosols. In particular, for the main aerosol

layer in the low atmosphere, updrafts transport aerosols in the main layer to places above
it. This reduces aerosol concentrations in the main layer, leading to reduction in radiative
heating of air by aerosols, CAPE, updrafts and cloud mass. This reduction enhances with
increasing aerosol concentrations in the main layer. For the aerosol layer in the upper
atmosphere, downdrafts transport aerosols in the layer to places below it. However, this
does not affect aerosol concentrations, radiative heating of air in the low atmosphere
significantly. This in turn does not affect CAPE and cloud mass significantly.

Aerosol radiative effects suppress clouds and reduce cloud mass by reducing solar

radiation which reaches the surface as compared to a situation when there are no aerosol
radiative effects. This suppression of clouds increases with increasing aerosol
concentrations in the main aerosol layer. Aerosol microphysical effects enhance cloud
mass and these effects are stronger with higher aerosol concentrations in the main layer.
When aerosol radiative effects, which are in terms of radiative heating of air by aerosols,
and aerosol microphysical effects work together, differences in cloud mass between a
situation where the main layer is in the upper atmosphere and that where the main layer is
in the low atmosphere enhance as compared to a situation where only aerosol



microphysical effects are present. More aerosols, and thus stronger radiative heating of air
and stronger aerosol microphysical effects in the main aerosol layer in the low atmosphere
enable this enhancement to be larger when aerosol concentrations are high in the main layer
than they are low.
This study shows that radiative heating of air by aerosols in the low atmosphere, which
are around or below cloud bases, enhances instability, invigorates convection and increases
cloud mass, which is contrary to the conventional wisdom of impacts of absorbing aerosols
on convection. However, radiative heating of air by aerosols in the upper atmosphere,
which are around or above cloud tops, enhances stability, suppresses convection and
reduces cloud mass. Aerosols in the low atmosphere intercept more solar radiation reaching
the surface, which tend to suppress the surface fluxes and convection, than aerosols in the
upper atmosphere. Here, we see that aerosol-induced changes in the surface fluxes and
those in radiative heating of air interact with each other in terms of responses of convection
and clouds to aerosols. This interaction varies with the varying vertical location of aerosols
and the varying cloud-induced wind that is at cloud scale. In general, traditional
parameterizations for warm cumulus clouds in climate and weather-forecast models have
not been able to consider this dependence of the interaction on the vertical location of
aerosols, since in general, those parameterizations for warm clouds do not differentiate
aerosol layers based on their vertical locations. In addition, the cloud-scale cloud-induced
wind, which is not able to be resolved by general resolutions in climate and weather-
forecast models, have not been represented by those parameterizations with good
confidence. So, impacts of aerosol transportation by cloud-induced wind on the interaction
have not been properly considered in those traditional parameterizations. Results here
demonstrate that for more comprehensive representation of interactions between warm
cumulus clouds and aerosols, we need to develop a more comprehensive parameterization
that is able well represent the varying interaction between aerosol-induced changes in the
surface fluxes and those in radiative heating of air with varying vertical locations of
aerosols and aerosol transportation by cloud-induced wind.







**Code/Data source and availability**

Our private computer system stores the code/data which are private and used in this study. Upon approval from funding sources, the data will be opened to the public. Projects related to this paper have not been finished, thus, the sources prevent the data from being open to the public currently. However, if information on the data is needed, contact the corresponding author Seoung Soo Lee (slee1247@umd.edu).

**Author contributions**

Essential initiative ideas are provided by SSL, JU and WJC to start this work. Simulation and observation data are analyzed by SSL, JU and KJH. CHJ. JG and YZ review the results and contribute to their improvement.

**Competing interests**

The authors declare that they have no conflict of interest.



**Acknowledgements**


This study is supported by the National Research Foundation of Korea (NRF) grant funded
by the Korea government (MSIT) (No. NRF2020R1A2C1003215 and No.
2020R1A2C1013278) and the "Construction of Ocean Research Stations and their
Application Studies" project, funded by the Ministry of Oceans and Fisheries, South Korea.
This study is also supported by Basic Science Research Program through the NRF funded
by the Ministry of Education (No. 2020R1A6A1A03044834).





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



**FIGURE CAPTIONS**

Figure 1. An inner rectangle in the map of the Korean Peninsula represents the simulation domain. The green (light blue) represents land (ocean) area in the map.

Figure 2. Vertical distributions of potential temperature and water-vapor mixing ratio at 09:00 LST on April 15th, 2015. These distributions are obtained from radiosonde sounding near the simulation domain in Figure 1.

Figure 3. Vertical distributions of the area-averaged aerosol concentrations at the first time step of (a) the control run and (b) the aro-above-cld run.

Figure 4. Vertical distributions of the time- and area-averaged cloud-liquid mass density that represents cloud mass for the standard simulations (i.e., the control, aro-above-cld, control-1500 and aro-above-cld-1500 runs).

Figure 5. Time series of the domain-averaged (a) liquid-water path, (b) updraft speed, (c) condensation rate and (d) convective available potential energy in the standard simulations.

Figure 6. Vertical distributions of the time- and area-averaged radiative heating rate (a) in the control and aro-above-cld runs over the initial stage between 10:00 and 13:50 LST, (b) in the control and aro-above-cld runs over the whole simulation period and (c) in the control-1500 and aro-above-cld-1500 runs over the whole simulation period.

Figure 7. Vertical distributions of the time- and area-averaged aerosol concentrations (a) in the control and control-novary runs, (b) aro-above-cld and aro-above-cld-novary runs, (c) control-1500 and control-novary-1500 runs and (d) aro-above-cld-1500 and aro-above-cld-novary-1500 runs.

Figure 8. Vertical distributions of the time- and area-averaged cloud-liquid mass density. In (a), the control-novary, aro-above-cld-novary, control-1500-novary and aro-above-cld-





1500-novary runs and in (b), the control-norad, aro-above-cld-norad, control-1500-norad
and aro-above-cld-1500-norad runs are shown together with the standard simulations.

































| Simulations | Altitudes of a main aerosol layer (km) | Aerosol concentrations in the main aerosol layer at the first time step (cm$^{-3}$) | Aerosol evolution | Aerosol radiative effects |
|---|---|---|---|---|
| Control | 0 - 1 | 15000 | Present | Present |
| Aro-above-cld | 2.5-3.5 | 15000 | Present | Present |
| Control-1500 | 0 - 1 | 1500 | Present | Present |
| Aro-above-cld-1500 | 2.5-3.5 | 1500 | Present | Present |
| Control-novary | 0 - 1 | 15000 | Absent | Present |
| Aro-above-cld-novary | 2.5-3.5 | 15000 | Absent | Present |
| Control-1500-novary | 0 - 1 | 1500 | Absent | Present |
| Aro-above-cld-1500-novary | 2.5-3.5 | 1500 | Absent | Present |
| Control-norad | 0 - 1 | 15000 | Present | Absent |
| Aro-above-cld-norad | 2.5-3.5 | 15000 | Present | Absent |
| Control-1500-norad | 0 - 1 | 1500 | Present | Absent |
| Aro-above-cld-1500-norad | 2.5-3.5 | 1500 | Present | Absent |


Table 1. Summary of simulations














| Simulations | Net solar radiation flux reaching the surface (W m$^{-2}$) | Surface latent heat fluxes (W m$^{-2}$) | Surface sensible heat fluxes (W m$^{-2}$) | Surface latent heat fluxes plus surface sensible heat fluxes (W m$^{-2}$) |
|---|---|---|---|---|
| Control | 293 (205) | 175 (120) | 22 (16) | 197 (136) |
| Aro-above-cld | 306 (217) | 170 (117) | 48 (33) | 218 (150) |
| Control-1500 | 461 | 250 | 70 | 320 |
| Aro-above-cld-1500 | 467 | 248 | 75 | 323 |


Table 2. The time- and area-averaged net solar radiation, latent heat, sensible heat and total
heat (sensible plus latent heat) fluxes at the surface over the whole simulation period in the
standard simulations. Numbers in the parentheses are averaged over the initial period
between 10:00 and 13:50 LST for control and aro-above-cld runs.



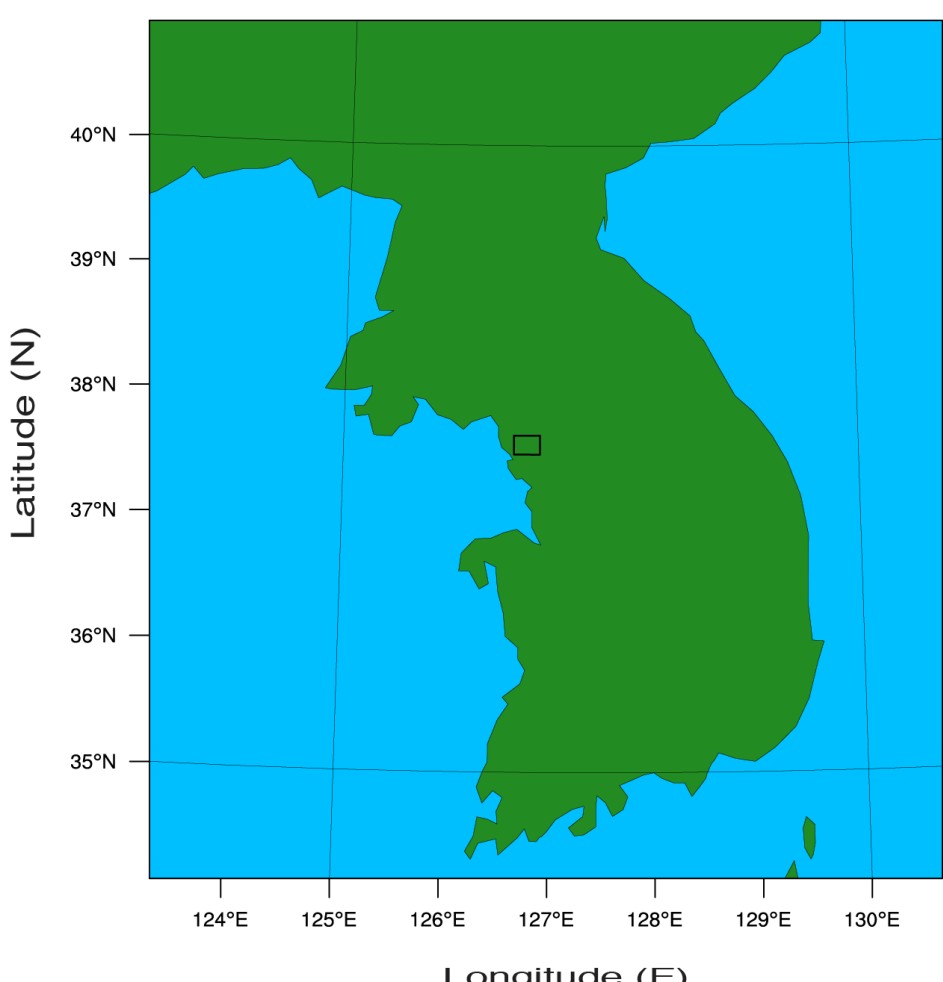


**Figure 1**











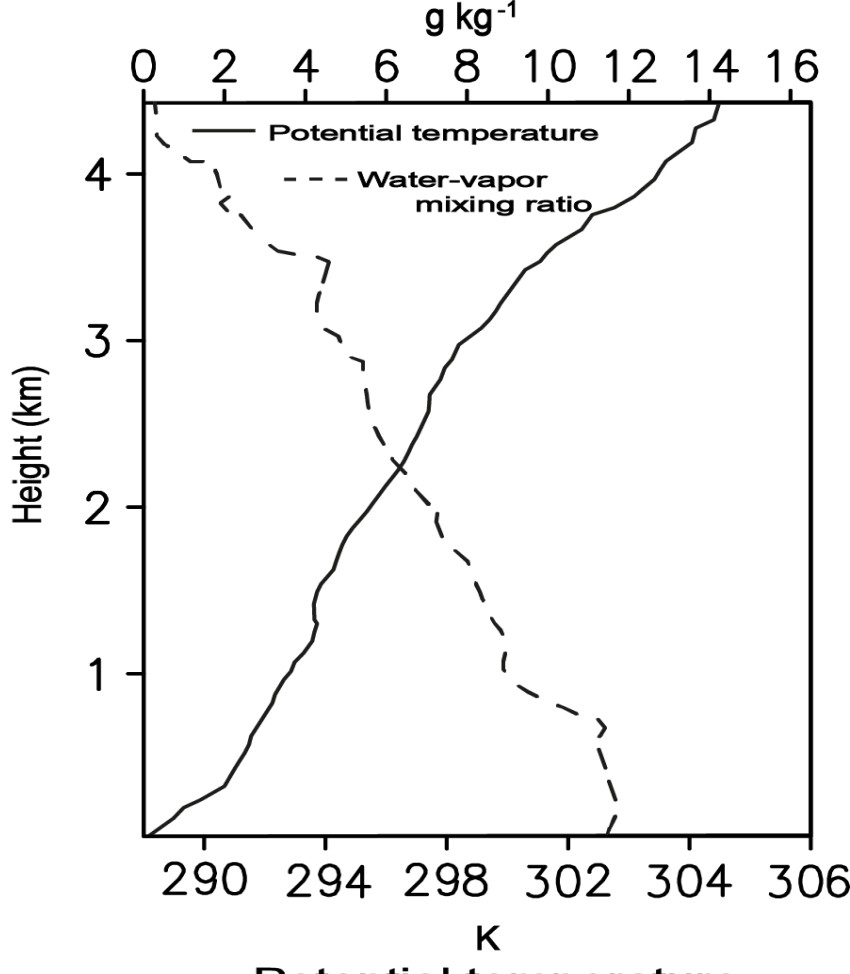


**Figure 2**









## Vericial distributions of aerosol concentrations

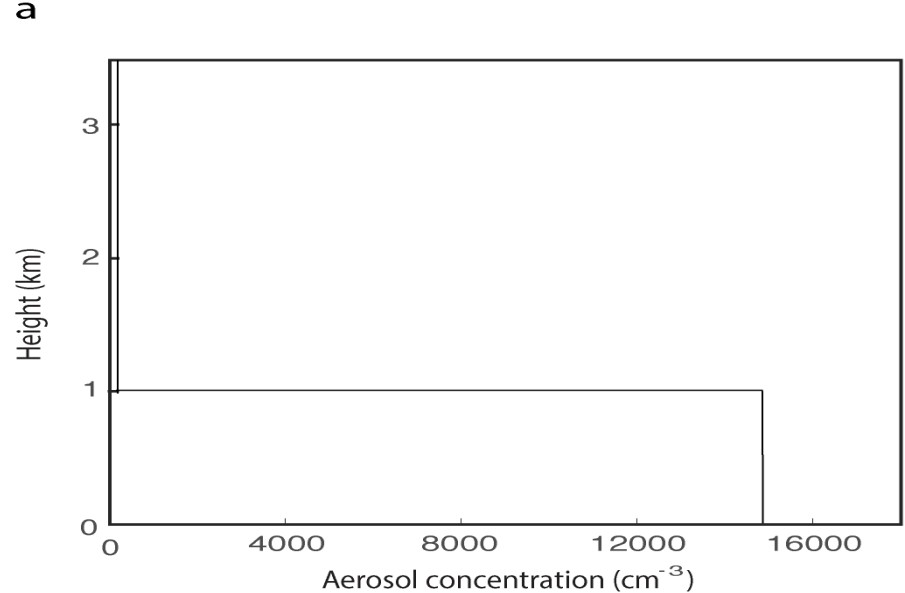

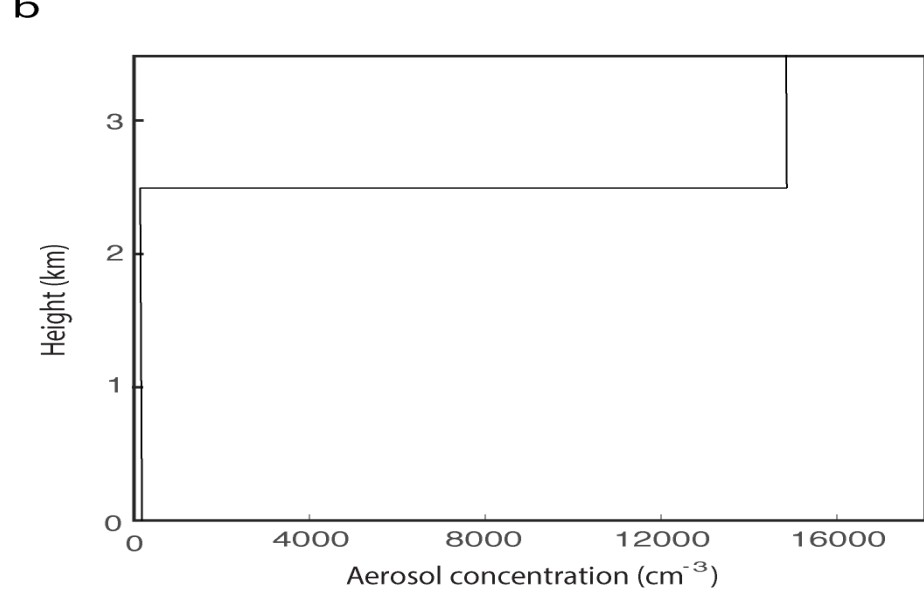


905                                    **Figure 3**






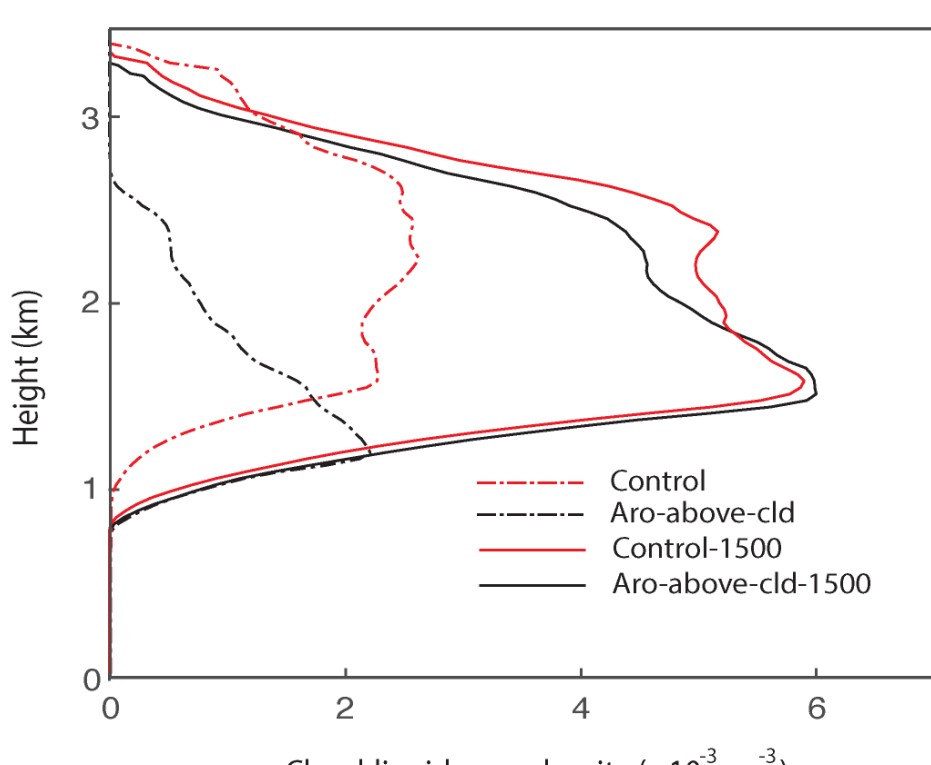


**Figure 4**











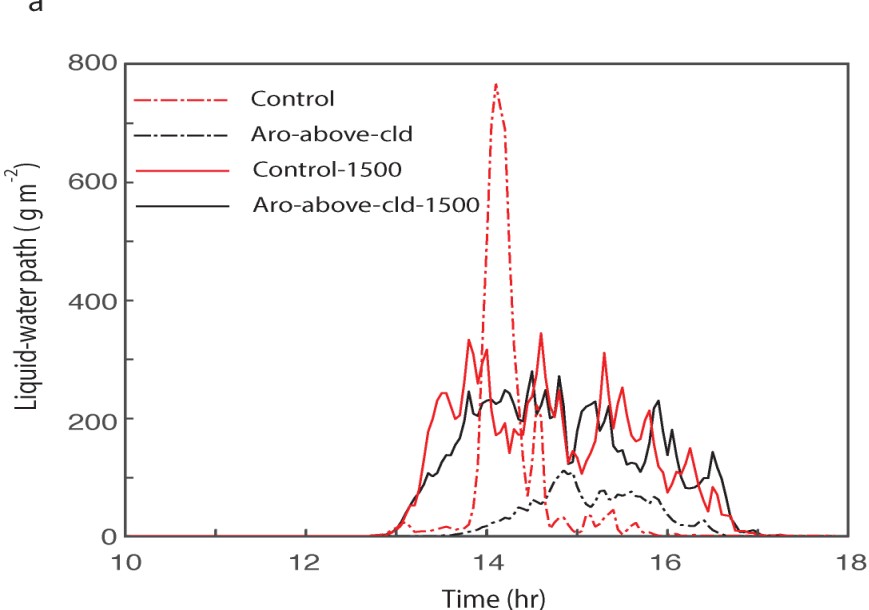

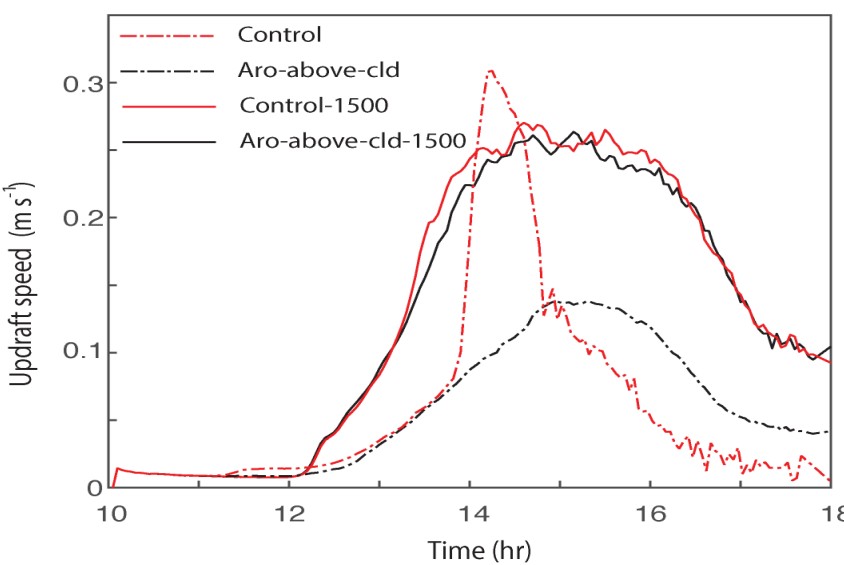


**Figure 5**







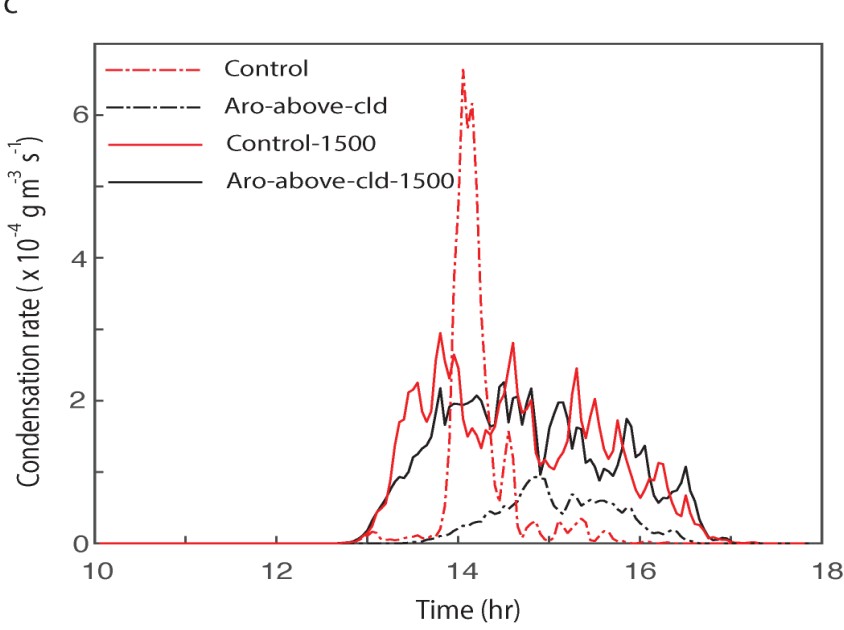

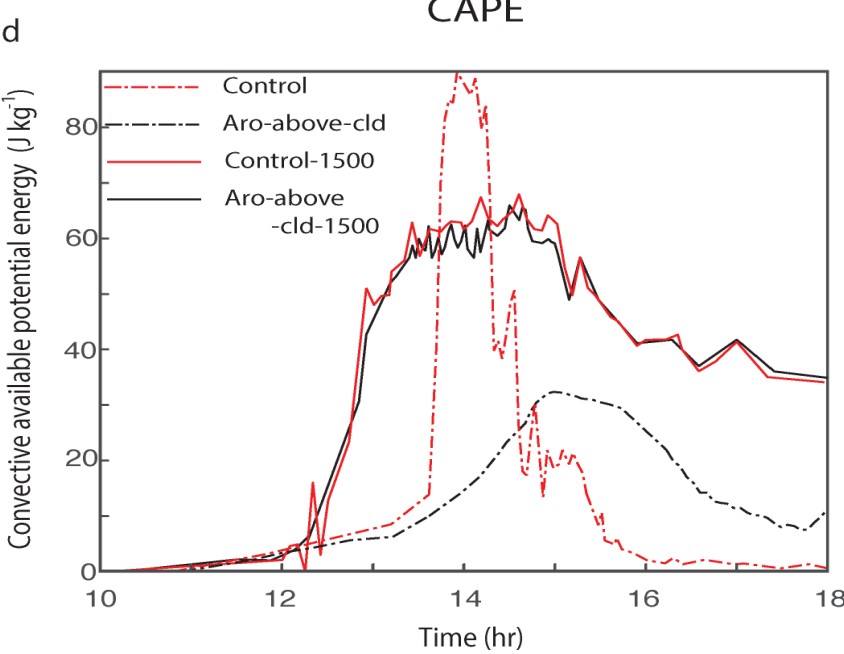

**Figure 5**



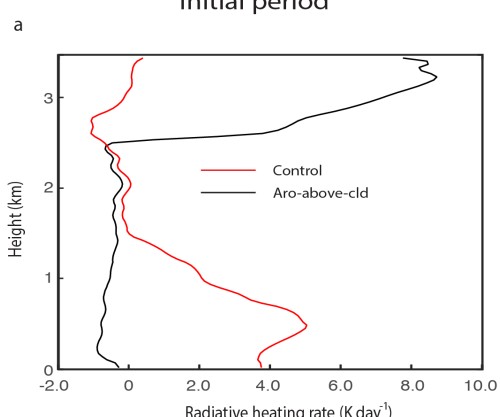

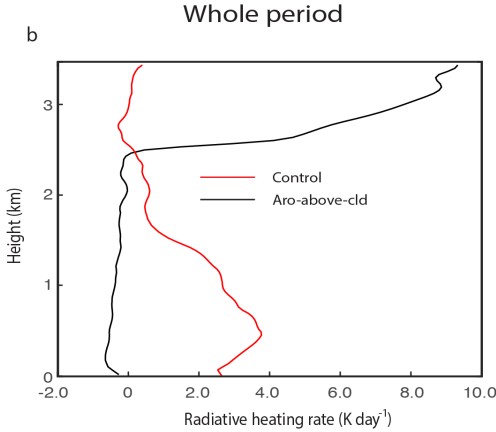

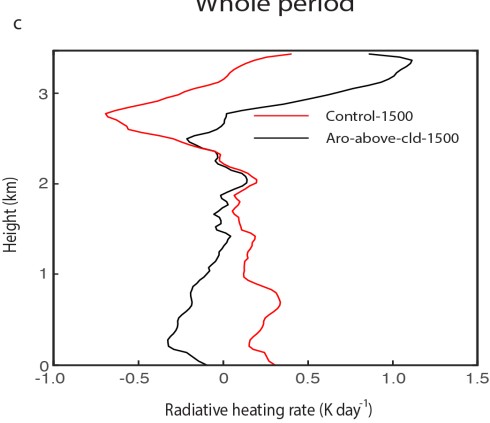


**Figure 6**



## Vertical distributions of aerosol concentrations

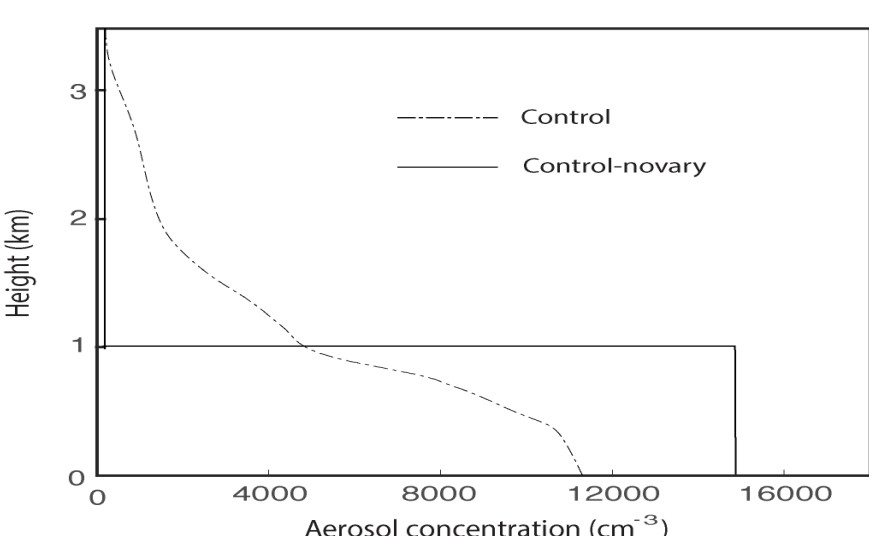

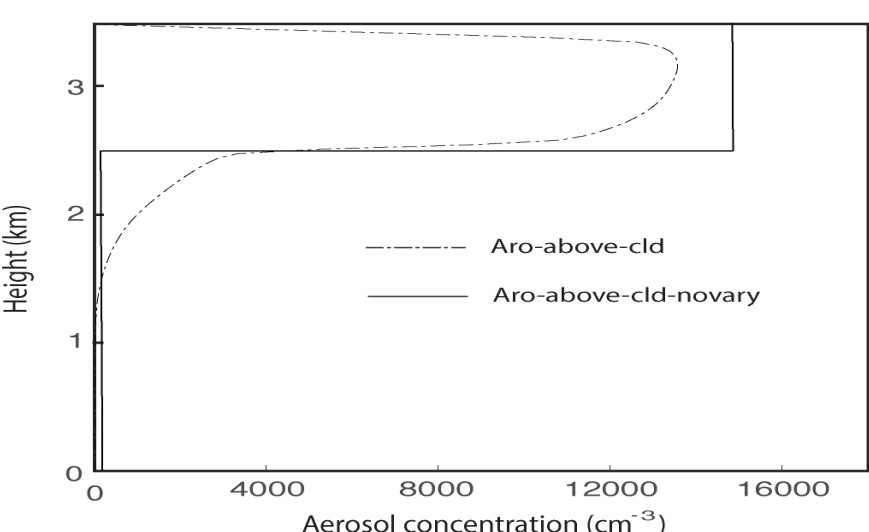


**Figure 7**



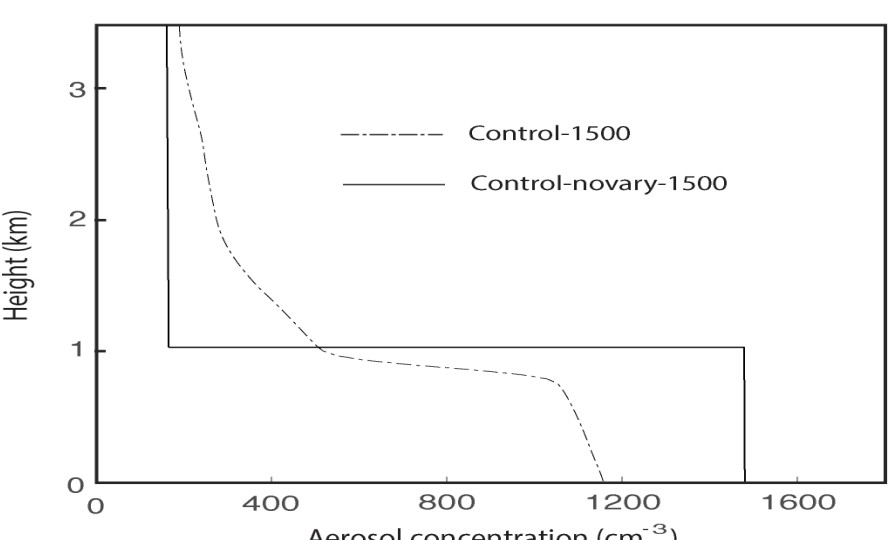

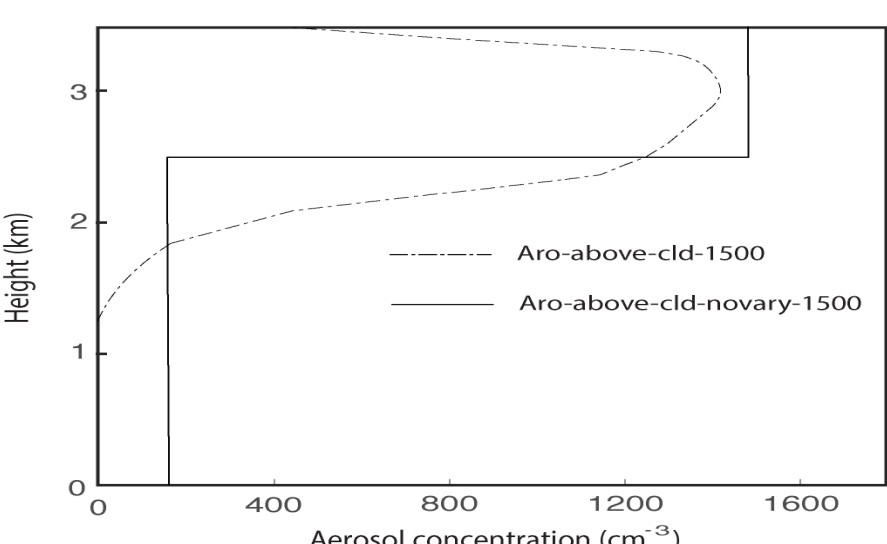

928        **Figure 7**




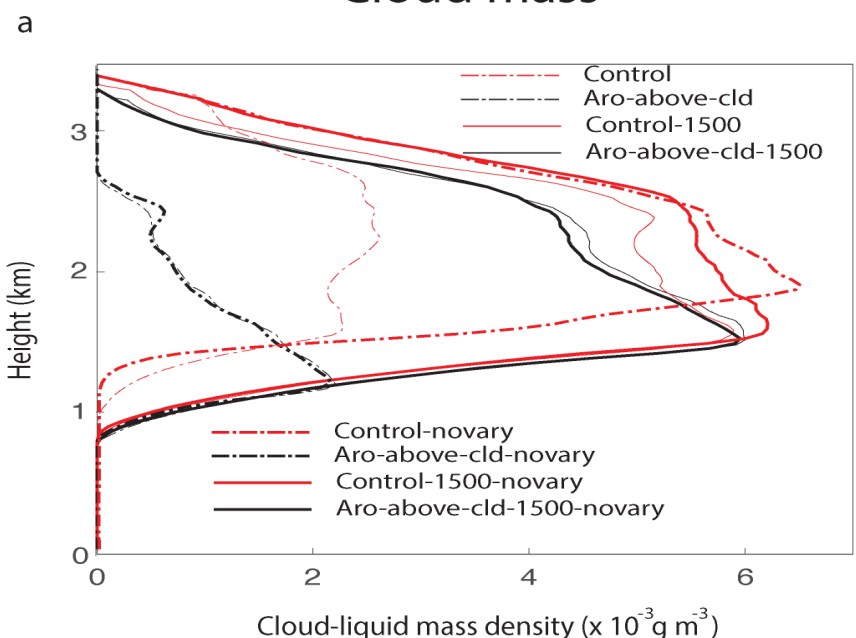

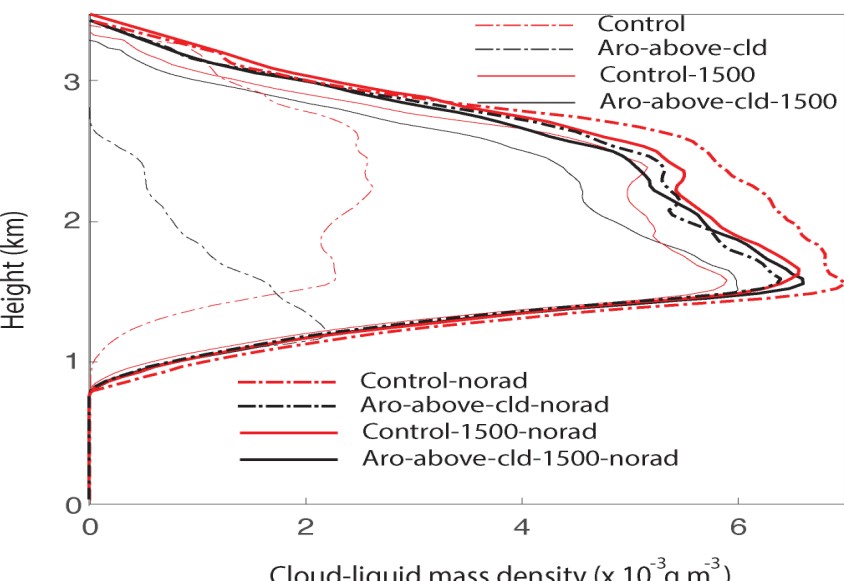


**Figure 8**