# Peer review of "Impacts of an aerosol layer on a mid-latitude continental system of cumulus clouds: how do these impacts depend on the vertical location of the aerosol layer?"

_Atmospheric Chemistry and Physics, 2022_

## Author Comment (AC1)

First of all, we appreciate the reviewer's comment. In response to it, we have made relevant revisions to the manuscript. Listed below are our answer and the changes made to the manuscript according to the comment. The comment of the reviewer (in black) is listed and followed by our response (in blue).

The paper describes the effects of the presence of aerosol layers above or below the cloud top/bottom by means of repeated simulations of a case study over the Korean peninsula. In addition, the study considers the effects of a different aerosol concentration in the layer, and separates the impacts of the radiative effects and the updrafts/downdrafts transport of the aerosols from the layer.

While the study points out relevant aspects in the interactions between clouds and aerosol layers, the manuscript is unfortunately hard to read and follow. The English in particular would need to be reviewed and the sentences restructured (I pointed out the most difficult to understand in the specific comments). Also, I found that some of the descriptive paragraphs on the different experiments were quite challenging to read. While the various simulations exercises are actually based on the simple concept of modifying one single feature at a time to see the effects, the description of the impacts are sometimes repetitive or diluted on a long dispersive text. I would suggest instead to shorten those sections and use, in all the various sections, the concise and straight-to-the-point style that the authors adopted on the paragraph between lines 435-452, which clearly emphasize the physical response of the added aerosol feature on the cloud system.

Following the comment here, the manuscript is revised substantially to make it concise and straightforward. For the revision, many paragraphs and expressions are restructured, and redundant text is removed. For the details of the revision, see the new manuscript.

To conclude, I would suggest a major revision of the manuscript before having it accepted for publication on ACP.

General comments:

Page 4, lines 99-109: As many important concepts are introduced in this paragraph, it would be beneficial to add some references.

References are added. See text between lines 94 and line 110 on page 4 in the new manuscript for details.

Page 5, lines 125-126: I would suggest to eliminate this sentence as the concept is repeated later on lines 130-132. The introduction mentions among the objectives the

improved understanding effects of a cloud deck on a cloud deck but the paper then focuses only on aerosols layer.

The introduction including the text pointed out here is revised substantially to make it succinct. This revision reflects points raised by the reviewer here. See the introduction in the new manuscript for details. In the new manuscript, the last paragraph in the introduction describes the aim of this study.

Page 6: It is not clear the time resolution at which the simulation is run. Same for the time length. I would assume that it corresponds to the event length (10:00 to 18:00 LST) but that's not clearly stated in the manuscript.

The corresponding text is revised as follows:

(LL172-173 on p6)

The cloud system is simulated for a period between 10:00 and 18:00 LST on April 13th, 2016. This period includes a time span over which the system exists.

(LL177-178 on p7)

The time step or temporal resolution is set at 0.1 second.

Page 7, lines 183:  The authors say that there is an observed aerosol system without showing it, It would be useful to have a reference or show the data indicating it.

The observed aerosol layer or system is advected from the East Asia. This advection of aerosol layers has been monitored by a station in the Yellow sea and stations in the simulation domain; these stations measure PM2.5 and this monitoring or observation of the aerosol-layer advection or the advected aerosol system has been done by comparing PM2.5 in the Yellow sea to that in stations in the simulation domain. This comparison of PM2.5 among the stations for the aerosol layer involved in this study is described in Figure 4 and associated text between lines 194 and 202 on p7.

Page 7, lines 188: what does the (20) stands for? Is it a repetition of the 20km resolution?

Here, to shorten the sentence by removing repetitive expression, the corresponding text is used where words in parentheses match each other following the writing convention. Hence, in the sentence "The length of the domain in the east-west (north-south) direction is 20 (20) km", "(north-south)" matches "(20)" and then via this matching, this sentence delivers two meanings:

1. The length of the domain in the east-west direction is 20 km
2. The length of the domain in the north-south direction is 20 km

However, to remove confusion, here, we revise the corresponding text as follows:

(LL174-175 on p6)

The length of the domain in both the east-west and north-south directions is 20 km.

Also, in the rest of the manuscript, expressions using above-mentioned parentheses to match words are revised by removing those parentheses.

Page 7, lines 195: on April 15[th] 2015, from a radiosonde sounding collected close to the domain (how close? It would be better if you also pinpoint the radiosonde location on the map of figure 1)

The radiosonde site is marked by a black dot in Figure 1a.

Page 7, lines 198: What is an "open lateral boundary condition"?

Open lateral boundary condition basically allows a simulated system to move out of and move into a domain of interest by wind. In this boundary condition, the system moves out of and move into the domain via its boundaries.

In open lateral boundary condition, once the system moved out of the domain, this system does not come back to the domain or affect variables in the domain. This emulates the advection of an observed system moving into and moving out of an area of interest in real nature. Once the observed advecting system moved out of the area of interest, this system does not come back to the area in real nature.

In the other boundary conditions such as periodic and symmetric lateral boundary conditions in the ARW model, a simulated system, which moved out of a domain of interest, does come back to the domain or affects variables in the domain. Hence, these boundary conditions do not reflect the situation in real nature. However, these boundary conditions are useful for idealized simulations. For example, in idealized simulations, periodic boundary conditions enable us to control the net budgets such as the net energy and momentum budgets in the domain in an efficient and simplified manner.

In summary, open boundary conditions may not be useful to constrain net budgets of variables such as energy and momentum unlike the periodic and symmetric boundary

conditions. However, open boundary conditions can simulate systems more realistically than periodic and symmetric boundary conditions.

The details of the boundary conditions in the ARW model can be found at *https://opensky.ucar.edu/islandora/object/technotes:479* and *http://dx.doi.org/10.5065/D6DZ069T*

Page 7, lines 200: Where is the AERONET site from which the data are used? Please indicate it also on the map of Figure 1.

The AERONET site is marked by a red dot in Figure 1b.

Page 7, lines 206-216: The authors here refer a lot to the AERONET data (chemical composition, size distribution, aerosol concentration) but none is shown, it will be useful to have some of the data plotted.

The size distribution based on the AERONET data is shown in Figure 5. Aerosol concentration is extracted from this size distribution, hence, by showing the distribution, information on aerosol concentration is plotted together.

Aerosol composition is described with percentage numbers in text. Maybe, we can plot a bar graph for these numbers. However, this bar graph is just about those numbers depicted as bars and does carry the same information as the numbers themselves. Hence, we consider the bar graph redundant and do not present it.

Page 7, lines 220: What does justify the choice of 150 cm-3 as a concentration above the layer?

Just before the aerosol layer is advected into the domain, the aerosol concentration is ~150 cm$^{-3}$ in the domain according to the AERONET measurement. This aerosol concentration is assumed to be background or environmental aerosol concentration that is not affected by the advected aerosol layer. Based on this assumption, aerosol concentration is set at 150 cm$^{-3}$ outside the layer during the simulation period.

The following is added:

(LL217-222 on p8)

At 06:00 LST, ~1 hour before the advected aerosol layer starts to be present, the AERONET-measured aerosol concentration is ~150 cm$^{-3}$ in the domain. This aerosol concentration is assumed to be a background aerosol concentration that is not

affected by the advected aerosol layer. Based on this assumption, the initial aerosol concentration is set at 150 cm$^{-3}$ outside the layer.

Pages 10 line 309: what does the (052022) mean?

It has nothing to do with text and is removed.

Page 18 lines 530-537: It is not clear to me where the effect on CDNC is shown.

Our understanding of this comment is that we did not show quantitative features of the CDNC variation between the control-1500-norad and the aro-above-cld-1500-norad runs, and the reviewer wants to see those features. Hence, we add the average CDNC in those runs in text and revise text pointed out here as follows:

(LL471-481 on p16)

Note that aerosol activation mainly occurs around cloud bases in the low atmosphere and more aerosols induce more activation for a given thermodynamic condition. Hence, there are more aerosol activation (or nucleation of droplets) and higher cloud droplet number concentration (CDNC) when the aerosol layer is in the low atmosphere than in the upper atmosphere. The averaged CDNC over grid points with non-zero CDNC and the whole simulation period is 532, 57, 131 and 53 cm$^{-3}$ in the control-norad, aro-above-cld-norad, control-1500-norad and the aro-above-cld-1500-norad runs, respectively. Droplets act as a source of condensation, since individual droplets provide their surface areas onto which water vapor condenses. Hence, higher CDNC induces more condensation and this in turn induces stronger updrafts and more cloud mass with the aerosol layer in the low atmosphere than in the upper atmosphere.

Specific comments:

Page 3, lines 63-64: For better readability purposes I suggest to rephrase the sentence as: "This is turn makes differences in cloud mass, which is larger when the layer is in the lower atmosphere and smaller when the layer is in the upper atmosphere"

Here, authors want to say that there is the variation of differences, which are between cloud mass when an aerosol layer is in the low atmosphere and that when an aerosol layer in the upper atmosphere, with varying aerosol concentrations in the aerosol layer, as seen in Figure 6 in the new manuscript.

The other reviewer suggests the following:

*I would concentrate less on trying to say every finding in the abstract and instead state a few things more clearly.*

Following the suggestion, abstract is revised substantially to make it succinct. Many sentences are shortened and during this process, some sentences are considered redundant and removed. The corresponding text pointed out here corresponds to the following in the new manuscript:

(LL60-63 on p3)

Hence, there is the variation of cloud mass with the location (or altitude) of the aerosol layer. It is found that this variation of cloud mass reduces, as aerosol concentrations in the layer decrease or aerosol impacts on radiation are absent.

Pages 3, lines 68-69: "Aerosol affects not only radiation but also aerosol activation" -> "Aerosol concentration affects…"

As mentioned in our response above, abstract is revised substantially to make it succinct. Many sentences are shortened and during this process, some sentences are considered redundant and removed. The corresponding text pointed out here is one of those sentences removed. See abstract in the new manuscript for the details of the revision.

Page 3, lines 71,74: Similarly as above, I suggest to shorten it as: "As aerosol impacts on radiation team up with those on the droplet nucleation, the cloud mass get larger when the aerosol layer is in the low atmosphere rather than when the layer is in the upper atmosphere"

Here, authors want to say that as seen in Figure 10b in the new manuscript, differences between cloud mass when an aerosol layer is in the low atmosphere and that when an aerosol layer in the upper atmosphere are greater when aerosol impacts on radiation and those on droplet nucleation both present than when aerosol impacts on radiation are present but those on droplet nucleation are absent.

As mentioned in our response above, abstract is revised substantially to make it succinct. Many sentences are shortened and during this process, some sentences are considered redundant and removed. The corresponding text pointed out here corresponds to the following in the new manuscript:

(LL60-63 on p3)

Hence, there is the variation of cloud mass with the location (or altitude) of the aerosol layer. It is found that this variation of cloud mass reduces, as aerosol concentrations in the layer decrease or aerosol impacts on radiation are absent.

Page 9 line 254: "roles of cloud impacts on aerosols in aerosol-layer impacts on clouds are identified" this sentence is not clear!

The corresponding text is revised as follows:

(LL252-254 on p9)

we aim to identify how cloud processes affect the aerosol layer and then the impacts of the layer on clouds.

Pages 10 line 288: I would remove the "As seen in" and start the sentence directly with "Figures 5b"..

The corresponding sentence is revised as follows:

(LL310-313 on p11)

Figures 7b and 7c show the time series of the domain-averaged updraft speed and condensation rates, respectively. These figures indicate that the average updraft mass fluxes and associated condensation rates in the control run are also slightly higher than in the aro-above-cld run for the period between 12:50 and 13:50 LST.

Pages 10 line 297-300: This sentence is not clear

The corresponding text is revised as follows:

(LL317-322 on p11)

Condensation is the only source of cloud mass in warm cumulus clouds. Also, updrafts with higher speeds tend to produce higher condensation rates for a given environmental condition. Hence, cloud mass, condensation rate and the updraft speed are closely linked to each other. This enables cloud mass, condensation rate and the updraft speed to be similar in terms of their temporal evolution in each of the control and aro-above-cld runs (Figures 7a, 7b and 7c).

Pages 10 line 310: "In Figure 5" or "In the panels of Figure 5"

"Figures 5" is typo and it is replaced with "Figure 7". Due to added figures, Figure 5 in the old manuscript becomes Figure 7 in the new manuscript. Here, "Figure 7" is equivalent to "the panels of Figure 7" following the reviewer and conventional interpretation.

Pages 12 lines 343-345: Please rephrase, this sentence is hard to read.

The corresponding sentence is revised as follows:

(LL358-361 on p12-13)

This more radiative heating in the low atmosphere during the initial period results in the subsequent jump in CAPE, associated higher CAPE, more intense updrafts and more cloud mass after the initial period by outweighing the lower surface heat fluxes in the control run.

Pages 12 line 357: I guess the authors are referring to the following hours, therefore I would rather say "the rest of the period"

We performed a substantial revision on text to make it more readable. During this process, the corresponding text is removed. See the last paragraph in Section 3.1 in the new manuscript for details.

Page 14 line 416: typo "conrol-1500 run"

Corrected.

Page 15 line 433: "The relatively short lifetime of the cloud system in the control run is shorter than.."  ->  "The lifetime of the cloud system in the control run is shorter than.."

The corresponding text and paragraph are revised substantially. See text between lines 410 and 421 on p14 for details.

Page 15 lines 446-448: I rather suggest "This means that with increasing concentrations of aerosols, the effects of radiative heating of aerosols in the low atmosphere enhances instability and cloud-liquid mass"

The corresponding paragraph is revised substantially to make it clearer. For this revision, to remove confusion, text pointed out here is removed. For details, see the paragraph between lines 398 and 409 on p14 in the new manuscript.

Page 16 lines 464-466: this sentence is a repetition of the lines 460-461.

The corresponding text is removed.

Page 18 line 523: "surface-reading"? Reaching?

Corrected to be "surface-reaching".

Page 20 lines 597-599: "this does not affect aerosol concentrations AND radiative heating of air.."

Done.

Figure 2, title: "Vericial" -> "Vertical"

It is found that Figure 3 in the old manuscript has this title with the typo. Following the comments by both of the reviewers, Figure 3 in the old manuscript is removed.

Figure 3: I do not find this figure relevant, it can actually be removed also because the same information is also reported on Figure 7.

Figure 3 is removed.

Figure 5 and 7: It would be better to have the whole 4 panels together in the same page

Done. See Figures 7 and 9 in the new manuscript.

---

## Author Comment (AC2)

First of all, we appreciate the reviewer's comment. In response to it, we have made relevant revisions to the manuscript. Listed below are our answer and the changes made to the manuscript according to the comment. The comment of the reviewer (in black) is listed and followed by our response (in blue).

The authors have performed a straightforward study to look at the differing impacts of aerosols on shallow convective clouds depending on the height of the aerosol layer. There are some worthwhile results, but a lot of the text and especially the results section is difficult to read. Parts of it have unnecessarily long sentences or are repetitive. I don't think this paper is ready for publication until a substantial rewrite has been done to make it clearer. A few specific comments follow.

Following the comment here, the manuscript is revised substantially. For the revision, many paragraphs and expressions are restructured, and redundant text is removed. For the details of the revision, see the new manuscript.

The abstract is rather cumbersome. For instance this phrase: "which are between a situation when the layer is in the low atmosphere and that when the layer is in the upper atmosphere" is used twice and I can't understand what it is saying. I would concentrate less on trying to say every finding in the abstract and instead state a few things more clearly.

Based on this comment, abstract is revised substantially to make it succinct. Many sentences are shortened and during this process, some sentences are considered redundant and removed. See abstract in the new manuscript for the details of the revision.

97 People *have started to*

Done.

98 clouds on clouds is a weird phrase to end a sentence on, I would reword this sentence

Reworded as follows:

(LL94-96 on p4)

In recent years, people have started to take interest in how aerosol layers affect clouds when these layers are above or around the tops of clouds (e.g., de Graaf et al., 2014; Xu et al., 2017).

Last paragraph of intro is repetitive and clunky.

The whole part of the introduction including the last paragraph is revised substantially to make it more succinct by removing redundant and repetitive text. See the revised introduction in the new manuscript for details.

144 is *used* for

Done.

Figure 1 - Rather than showing just a box on a blank map, maybe including a satelite image here would be good to set the stage for what kind of cloud scene this is.

Figure 2, which shows spatial distribution of satellite-observed cloud reflectivity in the simulation domain, is added.

215 Assumed is written twice

The corresponding text is revised as follows:

(LL207-210 on p8)

Based on the AERONET observation, the shape of the initial size distribution of aerosols acting as CCN is assumed to follow a bi-modal log-normal distribution as shown in Figure 5 in all parts of the domain.

Figure 3 is unnecessary - just state that the aerosol layer is between x and y km.

Figure 3 is removed and just the altitude of the aerosol layer is mentioned.

Do the results in the control run look at all like observations? Really any example of what the cloud field looks like would be helpful in interpreting the results.

The following is added to compare the control run to observation:

(LL274-298 on p10)

We utilize satellite and ground observations to evaluate the control run. The Moderate Resolution Imaging Spectroradiometer (MODIS) is a representative sensor on board polar-orbiting satellites. The MODIS passes the domain only at 10:30 am and 1:30 pm on each day. This means that it is difficult to get reliable data, which cover the whole simulation period, from the MODIS. The COMS, which is a geostationary satellite and

available in East Asia, does not provide reliable data of cloud mass. However, comparatively reliable data of cloud fraction and cloud-top height throughout the whole simulation period are obtained from the COMS. Data of cloud fraction and cloud-bottom height over the whole simulation period are collected from ground observations in the domain; note that ground stations which measure PM2.5 as marked in Figure 1b also measure cloud fraction and cloud-bottom height. Here, cloud fraction and cloud-bottom height in the control run are compared to those from ground observations. A comparison of cloud-top height is made in the domain between the control run and the COMS. Cloud fraction, which is averaged over all time points with non-zero cloud fraction over the whole simulation period, is 0.25 in the control run. Cloud fraction is 0.21 when it is averaged over all time points with non-zero cloud fraction that are collected from all ground stations in the domain over the whole simulation period. Cloud-bottom height, which is averaged over all air columns with non-zero cloud-bottom height over the whole simulation period, is 1.1 km in the control run.  Cloud-bottom height is 1.0 km, when it is averaged over all time points with non-zero cloud-bottom height that are collected from all ground stations in the domain over the whole simulation period. The average cloud-top height over all air columns with non-zero cloud-top height over the whole simulation period is 2.8 and 2.6 km in the control run and observation, respectively. The difference in each of cloud fraction, cloud-bottom and -top heights between the control run and observations is ~10%. This means that the control run is performed reasonably well.

Also, a satellite image of a cloud field as an example of the cloud field is added. See Figure 2 and associated text in the new manuscript.

505-507 An example, this could be rewritten more clearly as something like: "Figure 8b shows that with no aerosol radiative effects, the differences in cloud mass due to the height of the aerosol layer are much smaller."

Done.

**Citation**: https://doi.org/10.5194/acp-2022-385-RC2

---

## Author Response (AR2)

First of all, we appreciate the reviewer's comment. In response to it, we have made relevant revisions to the manuscript. Listed below are our answer and the changes made to the manuscript according to the comment. The comment of the reviewer (in black) is listed and followed by our response (in blue).

**Comments to the author**:
General comments

The paper focuses on the lower and middle troposphere with shallow convection and modeling of radiative aerosol cloud interactions over South Korea. This includes sensitivity studies for separating effects like altitude of the aerosol layer and aerosol amount, microphysics and radiation. Compared to the first version in ACPD it contains more information and is more concise.

Please use the meteorological conventions concerning altitude. The upper atmosphere usually refers to the stratosphere or mesosphere (i.e. at least above about 15km altitude) and not 4km which is the middle (or free) troposphere.

"Upper atmosphere" is replaced with "free atmosphere". Free atmosphere, instead of free troposphere, is inserted into the manuscript, since free atmosphere is more popular term than free troposphere and officially defined in the AMS glossary of meteorology.

At some places also atmosphere above or below the cloud layer might be used.

We inserted phrases "atmosphere around or below cloud bases" and "atmosphere around or above cloud tops" at proper locations of text. Here, "atmosphere around or below cloud bases" is equivalent to "atmosphere below the cloud layer", while "atmosphere around or above cloud tops" is equivalent to "atmosphere above the cloud layer",

I also wonder that "planetary boundary layer" is not used for the lowermost atmosphere.

"Low atmosphere" is replaced with "the planetary boundary layer"

The paper might be a useful contribution to ACP after further improvements.

Specific comments

Introduction: It should be mentioned that the focus is on cumulus clouds induced by shallow convection

To mention that this study is about warm cumulus clouds induced by shallow convection, the last paragraph in introduction is revised as follows:

(LL111-121 on p4-5)

Despite above-mentioned field campaigns, effects of aerosols above or around tops of warm cumulus clouds, which are induced by shallow convection, have not been examined as much as those of aerosols around or below bottoms of those clouds (Haywood and Shine, 1997; Johnson et al., 2004; McFarquhar and Wang, 2006). Motivated by this, this study delves into effects of not only aerosols around or below bottoms of warm cumulus clouds but also those above or around tops of those clouds. Through this, this study aims to contribute to the more comprehensive understanding of aerosol-radiation-cloud interactions. This more comprehensive understanding in turn contributes to more general parameterizations of those interactions for climate and weather-forecast models. To fulfill the aim, this study adopts the large-eddy simulation (LES) framework and an idealized setup for the aerosol layer.

Section 2.1: Please provide some information on horizontal and vertical resolution already here and not only partially in the next section.

The information on resolutions is moved from Section 2.2.1 to Section 2.1 as follows:

(LL127-130 on p5)

The Advanced Research Weather Research and Forecasting (ARW) model is used for LES simulations in this study. The ARW adopts a 50-m resolution for the horizontal domain. In the vertical domain, the resolution coarsens with height. The resolution in the vertical domain is 20 m just above the surface and 100 m at the model top.

Section 2.2.1: You may begin with: "As case study we simulate an observed system..."

Done.

Fourth paragraph of section 2.2.1 (before line 222) or Section 3: It would be nice to present a typical model result similar to the observations in Figure 2 (map of cloud reflectivity) for the standard control case. To present domain averages only is not sufficient in connection with weather forecast.

The simulated field of cloud reflectivity is added in Figure 6 and the related text is added as follows:

(LL269-273 on p10)

Figure 6 depicts the simulated field of the cloud reflectivity at 14:00 LST on April 13th, 2016 in the control run. Similar to the observed counterpart in Figure 2, simulated cloud cells are elongated in the southwest-northeast direction. Also, there is a good consistency in the overall cell size and population and the overall pattern of the spatial distribution of cloud cells between the observed and simulated fields.

Note that this added text is not in the fourth paragraph of Section 2.2.1 but in the first paragraph of Section 3.1 where the evaluation of the simulation is presented in the old and new manuscripts, since we believe that the added text is about the evaluation of the simulation.

Section 3.1, second paragraph: It would enhance the value of the paper if more information on the comparison between observations and model is provided in a figure or a table, maybe in a supplement if it is comprehensive.

More variables are compared between observations and the model, following this comment. Comparisons of these more variables and the previous variables in the old manuscript between observations and the model are shown in Table 3 in the new manuscript. Associated text is added as follows:

(LL273-277 on p10)

Table 3 shows comparisons of cloud and environmental variables between observation and the control run. Observation is performed by ground stations and satellites. Note that ground stations which measure $PM_{2.5}$ as marked in Figure 1b also measure cloud and environmental variables. Table 3 shows that differences in those variables between observation and the control run are ~10%.

Technical corrections

At plenty of lines including abstract and conclusions: please replace "upper atmosphere" by "atmosphere above cloud layer" or "free troposphere" or "middle troposphere" (see general comments)

"Upper atmosphere" is replaced with "free atmosphere"

Line 60: "a variation"

Done.

Line 459: you may shorten to "..effects and without"

Following this suggestion, the corresponding title is revised as follows:

(LL445 on p15)

Comparisons between simulations with and without aerosol radiative effects